SCIENCE FORUM

# Vision, challenges and opportunities for a Plant Cell Atlas

**Abstract** With growing populations and pressing environmental problems, future economies will be increasingly plant-based. Now is the time to reimagine plant science as a critical component of fundamental science, agriculture, environmental stewardship, energy, technology and healthcare. This effort requires a conceptual and technological framework to identify and map all cell types, and to comprehensively annotate the localization and organization of molecules at cellular and tissue levels. This framework, called the Plant Cell Atlas (PCA), will be critical for understanding and engineering plant development, physiology and environmental responses. A workshop was convened to discuss the purpose and utility of such an initiative, resulting in a roadmap that acknowledges the current knowledge gaps and technical challenges, and underscores how the PCA initiative can help to overcome them.

**PLANT CELL ATLAS CONSORTIUM, SURYATAPA GHOSH JHA[†], ALEXANDER T BOROWSKY[†], BENJAMIN J COLE[†], NOAH FAHLGREN[†], ANDREW FARMER[†], SHAO-SHAN CAROL HUANG[†], PURVA KARIA[†], MARC LIBAULT[†], NICHOLAS J PROVART[†], SELENA L RICE[†], MAITE SAURA-SANCHEZ[†], PINKY AGARWAL, AMIR H AHKAMI, CHRISTOPHER R ANDERTON, STEVEN P BRIGGS, JENNIFER AN BROPHY, PETER DENOLF, LUIGI F DI COSTANZO, MOISES EXPOSITO-ALONSO, STEFANIA GIACOMELLO, FABIO GOMEZ-CANO, KERSTIN KAUFMANN, DAE KWAN KO, SAGAR KUMAR, ANDREY V MALKOVSKIY, NAOMI NAKAYAMA, TOSHIHIRO OBATA, MARISA S OTEGUI, GERGO PALFALVI, ELSA H QUEZADA-RODRÍGUEZ, RAJVEER SINGH, R GLEN UHRIG, JAMIE WAESE, KLAAS VAN WIJK, R CLAY WRIGHT, DAVID W EHRHARDT*, KENNETH D BIRNBAUM* AND SEUNG Y RHEE***

**\*For correspondence:**
dehrhardt@carnegiescience.edu (DWE); kdb5@nyu.edu (KDB); srhee@carnegiescience.edu (SYR)

[†]These authors contributed equally to this work

**Competing interests:** The authors declare that no competing interests exist.

## Location-to-function knowledge can unlock new discoveries in plant science

The relationship between the sequence, structure and function of a protein has been explored from many angles, from biophysics to drug discovery (*Brausemann et al., 2017*; *Rajendran et al., 2018*; *Shang et al., 2020*). This long-standing paradigm has driven the development of innovative technologies in imaging, mass spectrometry, genomics and bioinformatics (*Camp et al., 2019*; *Mallis et al., 2020*; *Nakane et al., 2020*). Yet an evolving paradigm stating that the location of molecules is essential for their function at all scales, from molecular complexes to cells, organs, organisms and whole communities, also holds true (*Figure 1*).

At the molecular level, compelling examples of the location-to-function paradigm include the multi-enzyme complexes that work as molecular machines (so-called 'metabolons') to channel intermediate metabolites and produce specific molecular products in a highly controlled manner (*Obata, 2019*).

At the cellular level, the partitioning of proteins and lipids into organelles – and even

**Figure 1.** Location-to-function paradigm. The illustration shows the levels of organization of a plant (left) and a few examples highlight how location determines function at each level (right). At the molecular level, functions of protein complexes can be determined by their localization in membrane microdomains (***Jarsch et al., 2014***) or by dynamic protein interactions (***Obata, 2019***). At the cellular level, a protein can be located differentially through transport mechanisms, including vesicle trafficking (***Goring and Di Sansebastiano, 2018***) and nuclear translocation (***Marchive et al., 2013***), regulating its function. At the tissue level, cell position can drive its fate into a specialized cell type (***Shao and Dong, 2016***). Metabolic pathways can operate specifically in specialized cell types across tissues (***Marchive et al., 2013***; ***Schlüter and Weber, 2020***). At the next level, the existence of non-

*Figure 1 continued on next page*

*Figure 1 continued*

cell-autonomous transcription factors can transverse intercellular scales across plant organs (*Han et al., 2014*). Also, an organ-dependent post-translational proteome has been described as a mechanism of protein function regulation (*Uhrig et al., 2019*). At the organism level, plant interaction with biotic (*Harrison et al., 2002*) and abiotic factors (*Michaud et al., 2017*) can occur through a localized cue perception.

membrane microdomains within organelles – plays crucial roles in molecular trafficking and signal transduction, which have great impacts in many biological processes such as plant development and defence responses (*Jarsch et al., 2014*; *Xing et al., 2019*; *Yu et al., 2020*; *Heinze et al., 2020*; *Shimizu et al., 2021*). The differential location of a single molecule can determine one of several alternative functions (see, for example, *Smirnoff and Arnaud, 2019*). Protein function can also be dynamically regulated by a change in location, such as the activation of transcription factors by translocation into the nucleus (*Jiang et al., 2020*; *Marchive et al., 2013*; *Ramon et al., 2019*; *Pastor-Cantizano et al., 2020*). Furthermore, protein translocation to different cellular domains results in different functional outcomes (see for instance, *Kim et al., 2020*).

At the tissue level, $C_4$ photosynthesis is a classic example of how differential molecular localization between cell types enhances plant performance: differential expression of enzymes in bundle sheath and mesophyll cells in $C_4$ plants can concentrate $CO_2$ in close proximity to Rubisco, which is not possible in $C_3$ plants where the enzymes are present, but not differentially expressed between the cells (*Schlüter and Weber, 2020*).

Considering the opportunities in science and technology today, we contend that this 'location-to-function' paradigm will likely usher in conceptual leaps and transformative technologies, as was the case for the structure-to-function paradigm.

One particular area of potential success is agriculture, as the success of molecular approaches for crop improvement depends on a detailed understanding of the spatio-temporal regulation of genes and proteins and the production of metabolites (see *Wang et al., 2008* for an example of how controlling the location of gene expression impacts rice grain size and yield). The timing of expression is also a key determinant in imparting functional specification and influencing traits. For instance, engineered early expression of *WRINKLED1* (*WRI1*, an AP2 class transcription factor) during maturation increased *Arabidopsis* seed size and oil content (*Kanai et al., 2016*). More recently, an ambitious goal to globally sequester extra $CO_2$ into roots was established, which predicates on understanding what controls the development of a specific tissue, periderm, and the cell-type-specific expression of suberin, a lipophilic polymer found in plant cell walls. These examples illustrate how a thorough understanding of the cell- or tissue-specific expression of plant genes may

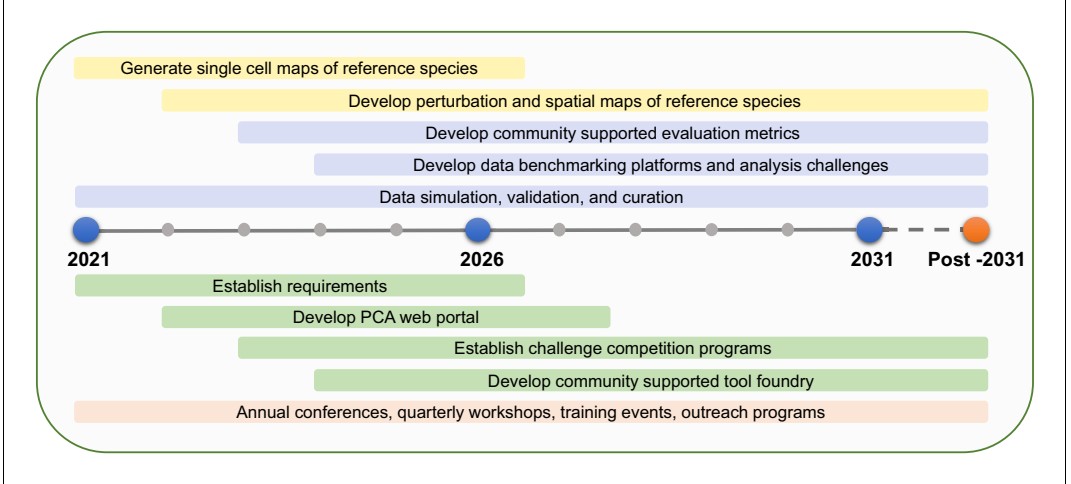

**Figure 2.** PCA milestones. PCA milestones for the next 10 years and beyond in data generation (yellow bars), data analysis (purple bars), software development (green bars), and building PCA community (orange bar).

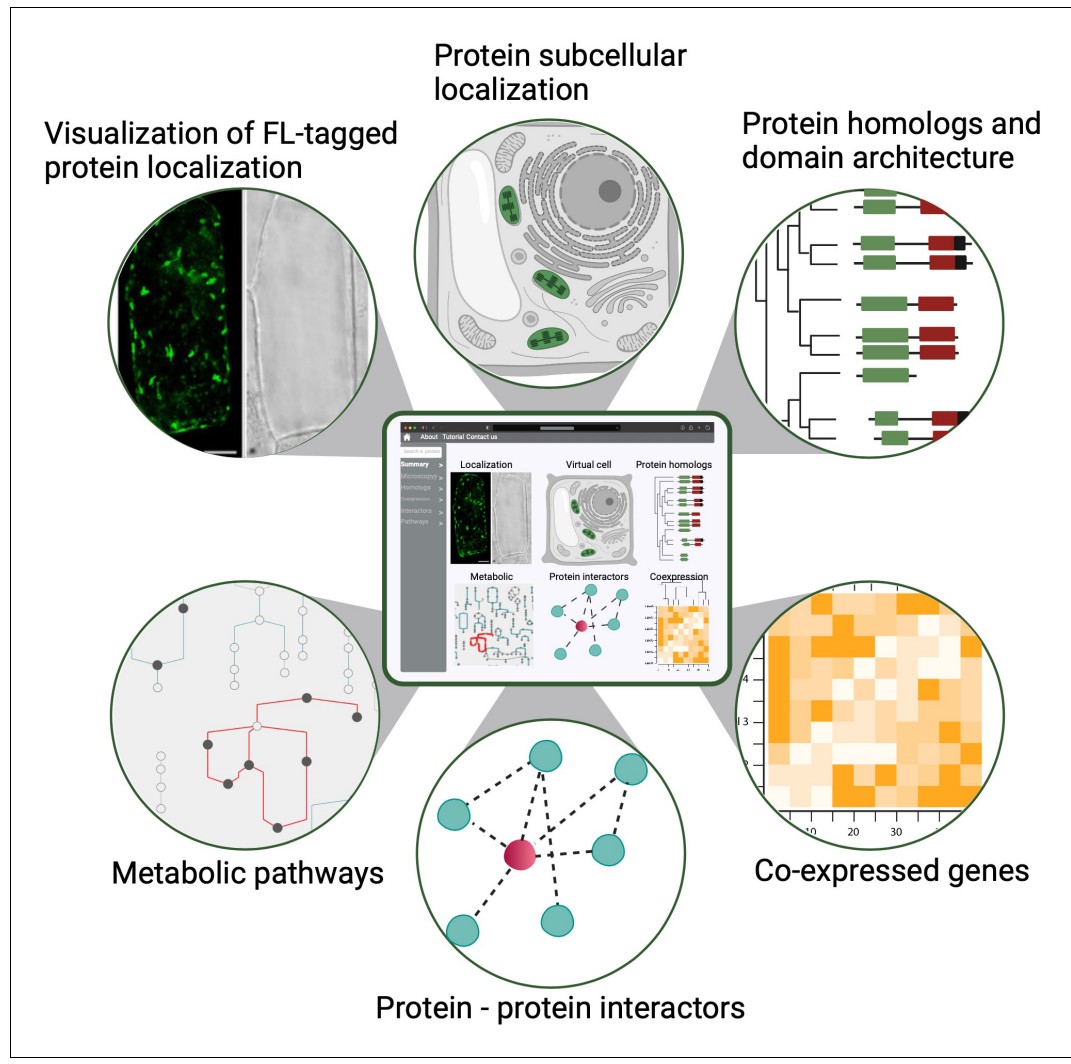

**Figure 3.** A conceptual diagram of a PCA user interface. An example of a PCA user interface that integrates various data types from molecular, biochemical, cellular and evolutionary contexts to connect location to function. Shown are example data types that would be seamlessly connected to enable easy navigation and discovery (FL: fluorescence).

enable the development of more effective biotechnological tools to facilitate crop improvement, plant resilience and climate change mitigation. Currently, these spatially- or cell-type-resolved expression data are missing or incomplete for most crop plants and there is no centralized repository for their integration. These shortcomings limit the rapid development of improved crops based on location-to-function data.

## Unlocking location-to-function: The first Plant Cell Atlas (PCA)

### The PCA workshop

Establishing a comprehensive picture of where molecules are located and how their locations inform function in higher plants will require concerted efforts from diverse groups of scientists. To catalyze such a community, a series of workshops were organized by Carnegie Institution for Science, New York University and Stanford University scientists around the topic of a cell atlas for plants, a framework to generate and synthesize molecular and cellular data on the development, dynamic functions and specialization of plant cells. This effort would also strive to

determine the state of the field, identify necessary components and people and discuss a path forward. Three workshops were held virtually during May and June of 2020. Over 400 scientists participated, of which 70% were early-career researchers, and 33% non-US based (*Rice et al., 2020*). Attendees of this first series of workshops became the basis of the PCA community. Following the success of the inaugural meeting, two technology-oriented workshops on single-cell sequencing and spatial proteomics and a career planning workshop were held virtually in 2021, respectively attracting 273, 253 and 42 participants. Additional workshops and symposia will be announced on the PCA website.

### The PCA goals

The goal of the PCA is to bridge gaps in knowledge, providing critical localization, dynamics and interaction information at the cellular and subcellular levels. The participants in the PCA Research Coordination Network envision a set of concrete milestones over the next decade to reach that vision, based on community goals set during these meetings in 2020 and 2021 (*Figure 2*).

1. Identify and quantify relative abundances of all definable cell types in every organ of several strategically selected reference species (e.g. a single-celled organism, a non-vascular plant, a eudicot, a monocot).
2. Profile the developmental trajectory of every specialized cell type in each organ in the reference species through single-cell sequencing.
3. Develop a 'reference' clearinghouse where new experiments based on single-cell profiling can be mapped onto comprehensive atlases representing the diversity of known transcriptional states among cell types.
4. Profile the responses of major organ systems to a common set of environmental perturbations at a single-cell resolution.
5. Determine the intraspecific variation of developmental trajectories and environmental responses at a single-cell resolution.
6. Establish data integration platforms and community-supported data standards to better enable data integration.
7. Develop a community-supported data benchmarking platform that will accomplish simulation and validation of data and evaluation metrics.
8. Determine protein subcellular localization, interactions and dynamics in the major cell types of leaves, roots and flowers for every gene in reference species.
9. Curate empirical datasets and develop data analysis and visualization tools through community-driven challenges (e.g., the DREAM single-cell transcriptomics challenge; *Alonso et al., 2019*; *Pham et al., 2020*).
10. Establish mechanisms for fostering new collaborations and funding for data generation, data integration as well as new technology and method development or adoption for plants.

Some of these goals can be accomplished quickly, as already existing technologies such as single-cell and single-nucleus RNA-seq make them realistic. Others require the development of technologies to collect information at the single-cell level. As technology and discovery reshape the vision of the PCA, new aspirations should emerge from the community. Nonetheless, these ambitious milestones represent key gaps in knowledge that will accelerate plant research (if addressed) and are feasible to achieve given the opportunities presented by recent technological advances. The PCA community can enable this vision in several ways.

### The PCA vision

First, the participants in the PCA Research Coordination Network envision a one-stop, user-friendly website with illustrated and graphical details about cellular organization across different developmental stages. This information, modeled after findable, accessible, interoperable and reusable (FAIR) data principles, could be widely accessed and used to advance research and develop educational tools (*Figure 3*; *Wilkinson et al., 2016*). Although single-cell genomics data (mostly transcriptomic) have been generated and analyzed in individual studies for several years, how well the broader community can use these datasets in comparative analyses has been limited. This is largely due to the difficulty in integrating and comparing data generated by different methods, growth conditions and collection methods. The PCA will be critical in harnessing the power of single-cell data by establishing and promoting community-supported best practices in sample preparation, data generation, analysis and integration – including data benchmarking platforms, evaluation metrics and reference cell maps for major

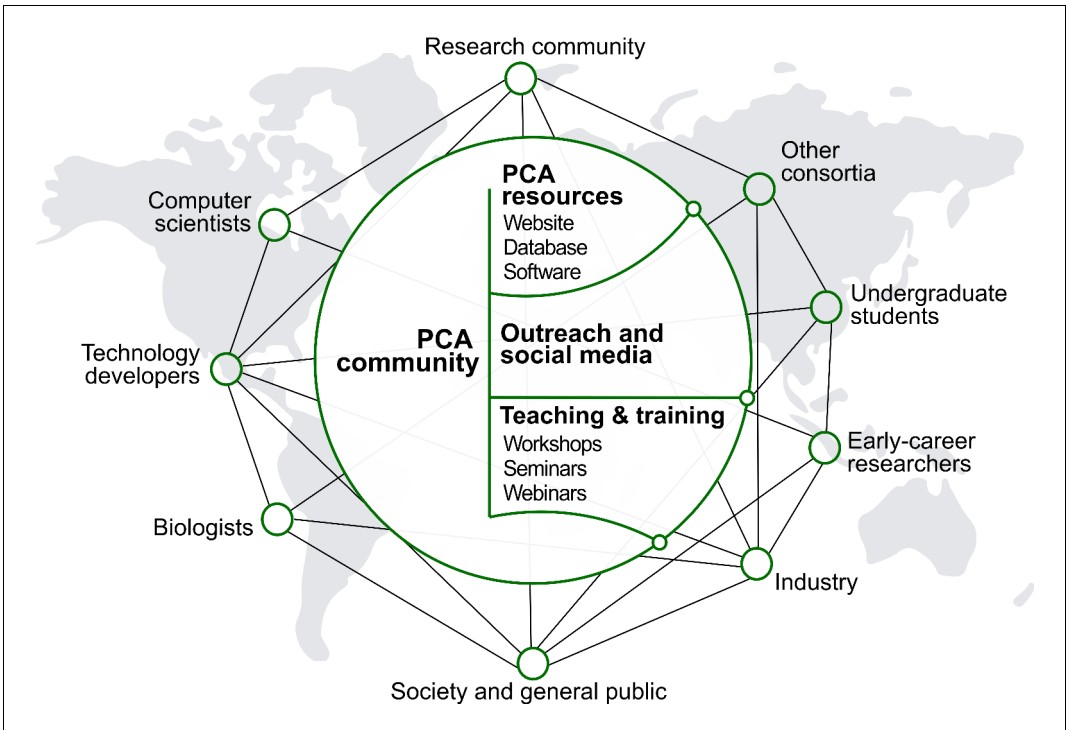

**Figure 4.** People and culture. Major stakeholders of the PCA are described. The goal is to establish a broad network of collaboration between developers and users through the creation of an accessible platform, educational tools and outreach activities. The PCA community should strive to be inclusive, diverse and transparent.

organ/cell types in model species. Important considerations for such infrastructure include how data will be gathered, curated and stored in order to integrate resources and contributions from different laboratories. It is necessary to develop and incentivize standards for both data collection (i.e. experimental protocols) and curation to allow integration of disparate datasets into a unified resource such as the PCA. This should include application programming interfaces (APIs) for programmatic access to ensure that PCA is accessible to all in a FAIR format. Similar standards have been produced for instance for plant phenotypic data (*Papoutsoglou et al., 2020*).

Second, we challenge and support the community to create a 4D representation of a developing plant from root to fruit (*Henkhaus et al., 2020*) with data collected from single-cell omics platforms (genomics, epigenomics, transcriptomics, proteomics and metabolomics) for each cell or cell type that are mapped onto tissue atlases of organ development. The data collected from the single-cell platforms form the basis of these atlases and serve as a reference for mapping further experimental data (e.g.

from perturbation experiments). The PCA efforts include integration of not only single-cell omics data, but also spatiotemporal dynamics of single cells and their connection to cell types and developmental trajectories. It therefore facilitates a comprehensive understanding of how different complex tissues and organs are formed during plant growth and development.

Third, we wish to generate excitement and awareness about the potential for plant science research to address society's most pressing challenges in agriculture, food security, bioenergy, resource management, as well as ecosystem stewardship and robustness in response to climate change. The PCA community needs to overcome the growing gap between ongoing research and societal perception with public communication and education. The PCA efforts should include outreach, training and education of the general public to understand the motivations, successes and limitations of this research. The initiative will help to build public awareness of how translational research that arises from model organisms can positively impact society and help feed the world. It will also demonstrate how understanding plants as foundational

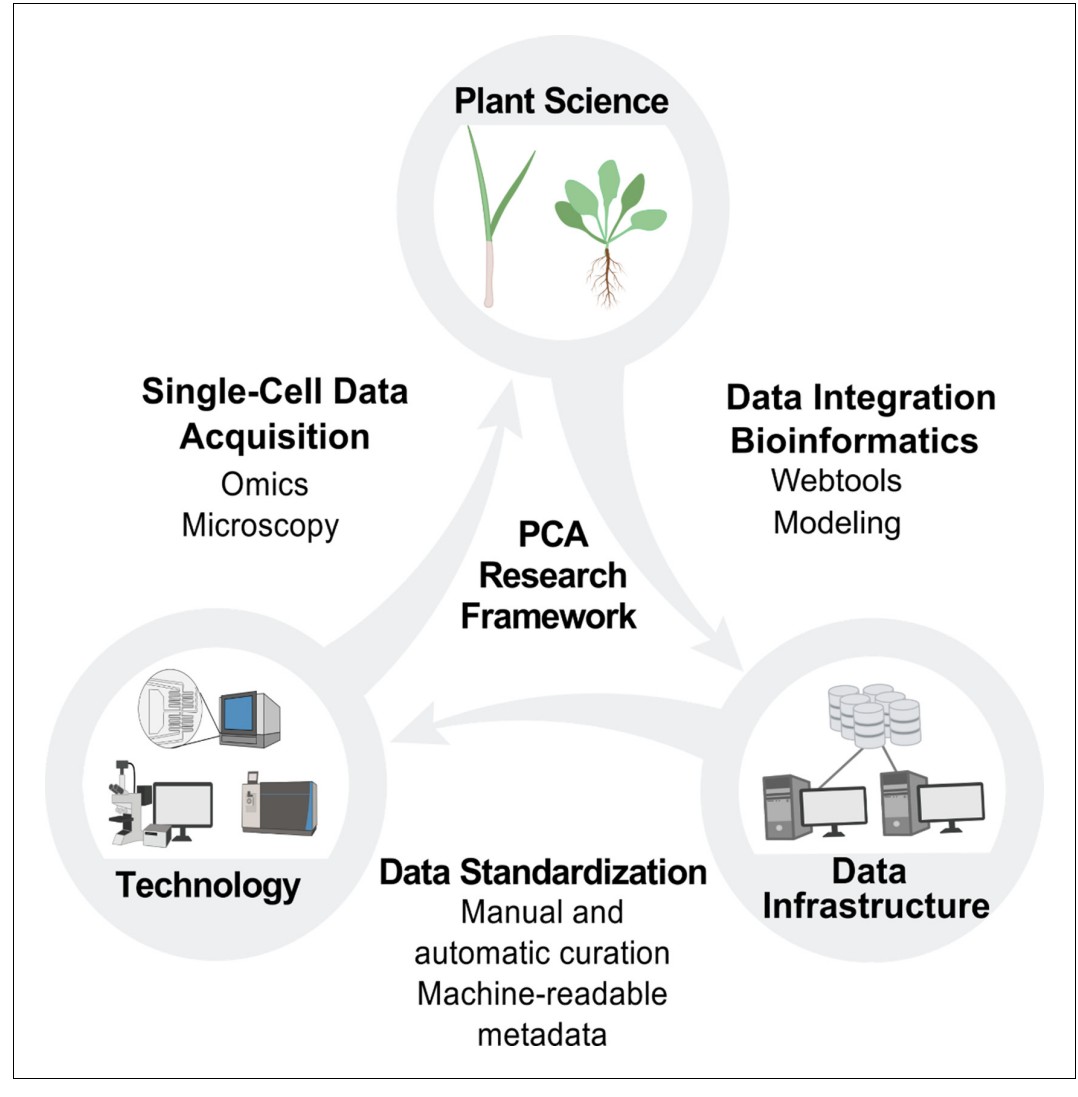

**Figure 5.** PCA research framework. Building the PCA will require cooperation and coordination between plant science, technology and data infrastructure. Plant scientists will use new technologies to generate data at the single-cell level, which will be integrated by data scientists into the PCA infrastructure through manual and automated curation. This data integration will enable the development of new technologies and drive new hypotheses and investigations for plant scientists, who will then contribute additional data to the PCA resource.

components of the ecosystem helps to predict and mitigate the consequences of climate change, and to be better stewards of the planet. Without these efforts, we fear that growing skepticism about science will impair the ability to aid society with the power of plant research, as apparent in the non-evidence-based regulation of gene editing in the European Union (*Jorasch, 2020*; *Ruffell, 2018*; *Zimny et al., 2019*).

Fourth, we see the PCA embracing new technologies and fostering multidisciplinary collaborations. The PCA will aim to build a community by introducing research tools to new users, and by facilitating interactions among scientists, engineers and breeders worldwide. These goals

can be met by organizing in-person events and online workshops with presentations from domain experts, followed by breakout rooms for targeted discussions and networking. The multidisciplinary nature of the PCA has enormous potential for accelerating the fundamental understanding of plant cells and how they connect to the whole organism, and for translating fundamental discoveries to agriculturally important crops across the globe. The dream of the PCA initiative in this regard is to actively engage applied plant scientists and breeders by facilitating the formation of joint national and international funding schemes across scientific disciplines. This will allow diverse communities

to benefit from each other's expertise and for ideas and technologies to be transferred from lab to land. Enhancement in research translatability will address globally relevant societal questions that are aligned with the UN Sustainable Development Goals (*Biermann et al., 2017*; *Costanza et al., 2016*; *Griggs et al., 2013*), including food sustainability (Goals # 1–3), resources to tackle climate change (Goal #13) and the overall survival of the biosphere (Goals # 14–15). We envision that starting with a comprehensive cell atlas of model organisms will help to better understand non-model species and eventually expand to an ecosystem-scale analysis of interactions between organisms.

### Inspiration from similar projects

What the participants in the PCA Research Coordination Network envisage is ambitious and not yet available for any system. However, examples of components of our vision will serve as guides. Among the animal whole organism atlases (*Lähnemann et al., 2020*), the mouse brain and adult mouse cell atlases are models for comprehensive coverage and community accessibility (*Rosenberg et al., 2018*; *Saunders et al., 2018*; *Zeisel et al., 2018*; *Tabula Muris Consortium, 2018*; *Han et al., 2018*). The Human Cell Atlas' data portal already includes data from 55 organs and 12 million cells, and it is becoming a model for making community-generated single-cell data available at a single portal, along with APIs and data analysis pipelines made easily accessible. In addition, the OpenWorm project provides an excellent example for modeling, visualizing and simulating various aspects of the biology and behavior of an organism (*Szigeti et al., 2014*). For plants, response to the environment would serve as an analogous level to behavior. EMBL-EBI's Single Cell Expression Atlas recently started hosting plant single-cell transcriptome data, which is an exciting development (*Papatheodorou et al., 2020*). Finally, the Global Natural Product Social Molecular Networking initiative provides a model for a one-stop-shop web infrastructure to enable data generation, integration, analysis and publication from mass spectrometry (*Wang et al., 2016*).

## The envisioned PCA community

The PCA will be developed, maintained and used by a global community with diverse scientific, technical, creative and educational backgrounds (*Figure 4*). Helped by rapidly evolving online tools that increase inclusivity and

transparency, the initiative should seek ways to facilitate the engagement, communication and collective decision-making among all groups involved. This work will aim to be grounded in active listening, psychological safety, teamwork, mutual respect and integrity. An inclusive and bottom-up ethos will underpin the development of global biological databases built with scientific rigor and transparency.

The participants in the PCA Research Coordination Network anticipate that the success of the PCA will hinge on an open and frequent dialogue between developers and users, with facilitation and guidance from a steering group. Recruiting scientists from diverse disciplines as well as from different career stages will be crucial in the successful development of the PCA. In particular, the initiative should continue to provide an opportunity for early-career researchers and foster the development of future leaders in plant biology.

### Developers

The mission of the PCA community will be achieved by groups who will contribute to the development and maintenance of the database, as well as those who generate or use the data and provide inputs for improvement. Developing the PCA database will be done with the expertise of scientists, software engineers, modelers, scientific illustrators and animators coming together to fine-tune the display of cellular organization and collecting information. Data and computational scientists will have to be involved in the curation and annotation of data. Engagement will extend to consortia pursuing similar activities in different biological systems (e.g., Human Cell Atlas). As the PCA will heavily depend on data visualization, user interface and experience design will also benefit from advisors in art and design fields. Additionally, collaboration with cartographers will enhance the understanding of spatial mapping for developing the PCA database. These influences from outside the field of plant research will complement and bolster the expertise of plant scientists in the PCA community.

### Users

The participants in the PCA Research Coordination Network expect several types of users. One of the major user groups will be the data generators, which will involve plant scientists from many disciplines, including single-cell profiling, protein-protein interactions, cell and

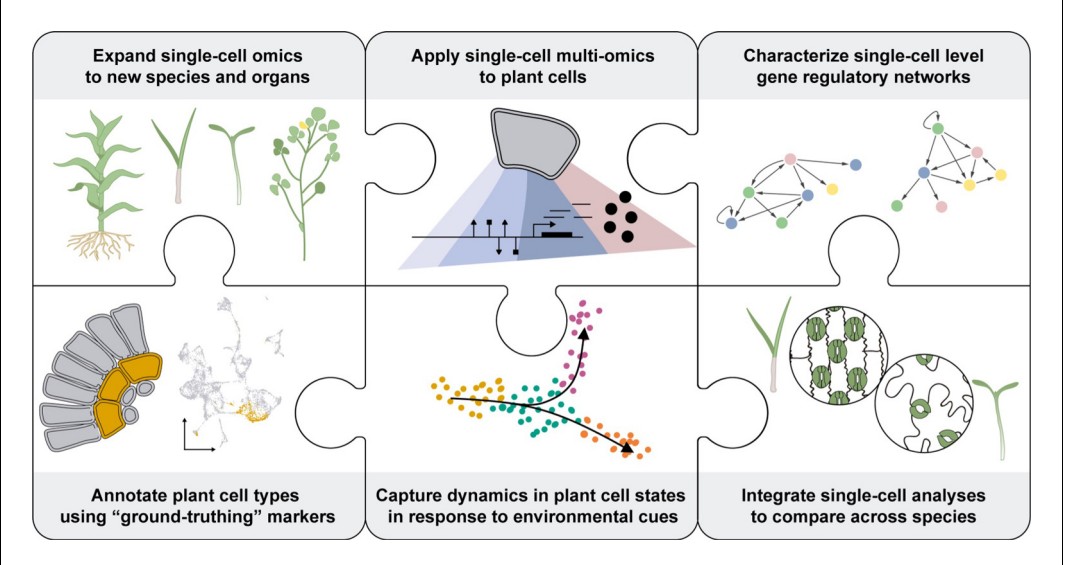

**Figure 6.** Knowledge gaps to fill for the PCA. Several limitations inherent to plants (shown as pieces of a jigsaw puzzle) must be overcome to enhance the understanding of plant cells as biological systems. To characterize the unique molecular attributes of each cell type composing a plant, plant biologists will need to develop reliable methods to broadly access the different cell types across various plant species. Also, single-cell multi-omics technologies will be necessary to get a deeper understanding of plant cell processes across all layers of molecular regulation. This data must be gained in the context of functional annotation of the cells. This would require the identification of reliable marker genes for each type of plant cell-. Ultimately, the spatiotemporal distribution of the molecular attributes of each cell will help to understand their dynamic regulation during plant cell development and in response to environmental stresses. Integrating the information collected using bioinformatics tools will enable the characterization of regulatory networks at a single-cell resolution and comparative analyses across plant species at the cellular level. This data will fuel the establishment of new synthetic biology strategies to enhance plant biology.

developmental biology, imaging or synthetic biology. To engage scientists from different career stages and areas of plant science, it will be important to develop mechanisms attributing and giving credit to data contributors. Another type of users will be researchers who wish to explore the PCA data to make discoveries. Engaging these users through a bottom-up process driven by their needs will be required for a vital and thriving PCA. Finally, the third group of users, both within and outside of the research community, are those who will access the PCA for educational purposes. Simple and intuitive interfaces combined with an attractive display of information and compelling visualizations will attract a large number and range of educators seeking information for teaching.

### *Integrating the PCA in the wider bioinformatics community*

Robust computational infrastructure development will be essential to support the research, community-building and outreach activities of the PCA (*Figure 5*). We envision the PCA infrastructure (that is, the technologies, standards and computational platforms used by developers to build a framework to house and make available the PCA-relevant data) to have a seamless integration of multi-omics data with localization data at multiple scales of the plant. A governance structure, contribution guide and code of conduct for participants in infrastructure development should be established to define and support community values. These values include open-source development, FAIR principles and outcomes-based planning and implementation. Stakeholders should be included in the decision-making process. To implement these values, existing efforts with similar values and goals should be leveraged. For example, the DREAM challenges group could be leveraged for developing data analysis, visualization and access tools (*Pham et al., 2020*).

While new tools and resources will be needed to build a PCA community platform, existing resources should be strong guiding principles when developing the PCA infrastructure. The PCA should make efforts to liaison with data

repositories, knowledge bases and data analysis hubs such as ATTED, eFP Browser, ePlant, Ensembl Plants, Gramene, MaizeGDB, TAIR, The 1001 Arabidopsis Genomes phenotype database, NCBI's GEO, EMBL-EBI's Expression Atlas, AraPheno and others (*Obayashi et al., 2007*; *Sullivan et al., 2019*; *Howe et al., 2020*; *Tello-Ruiz et al., 2021*; *Portwood, 2018*; *Berardini et al., 2015*; *Seren et al., 2016*; *Barrett et al., 2005*; *Papatheodorou et al., 2020*; *Waese et al., 2017*). For example, OMERO is a global consortium effort aiming to standardize biological imaging data, which includes data depository as well as image processing software to translate image files into a universal format. Cross-species information may be mapped to the latest phylogenetic tree updated by the Tree of Life Web Project and Bioschemas. Engagement should also extend to consortia that are undertaking related work in other organisms, such as the Human Cell Atlas. To create a collaborative research environment, projects such as Synapse, Cyverse and KBase could be evaluated for strategic partnership. To develop a framework to represent the PCA at multiple scales, it will be important to implement conceptual frameworks such as the common coordinate framework (*Rood et al., 2019*), developmental trajectories ('pseudo-time') and spatial gradients ('pseudo-space'; *Stewart et al., 2019*).

### Driving the community forward

The PCA will receive diverse sets of data from different types of techniques and experiments, many of which are constantly pushing the boundaries of the state of the art. As such, the PCA needs to position itself and its community at the frontier of plant biology, looking out for emerging findings and technologies. This could be achieved through online workshops that enable pioneering researchers and developers to introduce emerging technologies to the community. Other training activities might include hands-on workshops on single-cell, spatially resolved experimental techniques, data annotation and analysis tutorials, as well as hackathons and virtual office hours to help novice code developers, data scientists and other researchers.

## Gaps in knowledge: How the PCA could address unresolved questions in plant science

Many unresolved questions in plant biology require new technologies and approaches that the PCA could drive forward, as well as benefit from. Examples of such major knowledge gaps are described here (see also *Figure 6*).

### Plant development

The PCA could unlock answers to questions that require understanding the molecular changes in cells during development. The potential to characterize embryonic development comprehensively at single-cell resolution and in the multicellular context of complex tissues could help decipher the 'rules' underlying the earliest stages of plant tissue formation (*ten Hove et al., 2015*). In post-embryonic development, plant meristems are remarkable multicellular machines, continually producing new cells that must be specified to fate and then guided through phases of maturation (*Esau, 1953*; *Sinnott, 1960*). A major challenge is to understand the mechanisms that coordinate these cell maturation and fate specification events within meristems. Furthermore, plants present a fascinating contrast to mammalian systems with respect to the plasticity they maintain (*Borges, 2008*). For example, perennial plants often show meristematic plasticity, where only a subset of meristems per season are activated, and with the meristematic activity frequently modulated in response to external and internal cues (*Tylewicz et al., 2018*). The power of single-cell omics to capture the dynamic molecular coordination of tissue maturation at fine scales promises to help advance these classic questions in plant development (*McFaline-Figueroa et al., 2020*).

### Response to the environment

As for problems in development, how plants respond to environmental stimuli requires understanding the molecular changes taking place in individual cells and how these changes are communicated cell to cell, across tissues and among organisms. An example can be found in how the threat of invasion of foreign biota into cellular and intercellular spaces often induces localized hypersensitive responses of programmed cell death (*Jones and Dangl, 2006*; *Ngou et al., 2021*). The question remains as to how the signals induced at the attack sites are communicated to neighboring cells and propagated to

## Box 1. Major challenges in building a PCA.

1. Isolating plant cells and subcellular compartments. Efficient strategies for dissociating plant cells away from their structural context are needed to understand biological processes at the single-cell level. The fidelity and utility of methods requires extensive validation for collecting single-cell information across tissues and species.

2. Lack of well-characterized, cell-type-specific marker genes in non-model species. The functional characterization of plant cell types based on their transcriptomes suffers from the small pool of plant marker genes. The major limitation of the most widely applicable technology (Visium, 10x Genomics) is its relatively coarse resolution that would obscure the boundaries between several cell layers in plant tissues.

3. Incomplete annotation of plant genomes. Approaches that account for gene duplication, gene loss, and the potential for functional swapping among paralogs are required in plants. Comparison of individual transcript isoforms is difficult with current technologies. Efforts are needed to develop long-read technology to single-cell applications, or other strategies that can provide sequencing coverage to whole transcripts.

4. Need to integrate multiple components of the cell, such as nucleic acids (DNAs and RNAs), proteins, lipids, and metabolites. Addressing this challenge requires the development of multi-omics capabilities at cellular resolution, coupled with computational approaches that can integrate the data (*Libault et al., 2017*).

5. Plasticity of plant phenotypes. Plant phenotypes can be highly plastic according to changes in the environment. Efforts to establish reference atlases should ensure that standards in metadata description for growth conditions are established and adhered to.

6. Need to develop innovative strategies to functionally characterize plant genes at the single-cell level. Single-cell genetic technologies should be adapted to plant cells for high-throughput characterization of gene function in specific cells and cell types. For example, combining CRISPR-Cas9 technology with cell-type-specific inducible systems enabled the creation of genetic mosaics (*Decaestecker et al., 2019*; *Wang et al., 2020*). In addition, studies of proteins are limited to the whole plant-based tissue analyses. Approaches are needed for investigating single-cell or cell-type-specific proteomics in a high-throughput manner (*Balasubramanian et al., 2021*).

7. Limited resources. The plant science community is small compared to its animal or human counterparts and having access to more resources is a long-term challenge. Despite this, this community is relatively open and collaborative, making efforts to standardize aspects of single-cell experimental and analytical strategies potentially an easier barrier to surmount.

distal regions. Do hidden pre-patterned states among seemingly homogenous cells help the plant anticipate attacks? Or is there heterogeneity in cellular responses that could reveal a more complex response to infection? How do these processes differ between susceptible and resistant plant varieties? Heterogeneous responses to infection around invasion sites, as well as differences in molecular signals that correlate with cellular immunity, could provide new insights into how successful defense responses are coordinated.

A parallel set of questions emerge from biotic interactions involved in symbioses, such as the ancient association of land plants with mycorrhizal fungi, or more recent symbioses like those formed by members of the nitrogen-fixing Fabids clade (*Wang and Qiu, 2006*). In the latter case, infection by the symbiont through the root hairs takes place asynchronously, so bulk sampling often leads to profiles that are a heterogeneous mixture of cell states (*Libault et al., 2010*). Can new technologies help to gain a discrete view of how individual cells respond to the symbiont over time? Can fresh insights be gained into plant-symbiont interactions by studying root cells and specialized symbiont structures (i.e., arbuscules, nodules) simultaneously? In addition, can the chain of cellular communication in the plant throughout the stages of microbial colonization be better understood? Lastly, by comparing these processes across plants with differential symbiotic specificities, is it possible to start understanding the hallmarks of a robust symbiotic relationship?

## Cellular evolution

From rhizoids to pollen, plants have evolved many new specialized cell types, allowing adaptation to new ecological niches. In addition, specific cell types in the root harbor molecular mechanisms to perceive and respond to environmental stimuli, suggesting they may act as 'gatekeepers' for these responses (*Henderson and Gilliham, 2015*). What new perspectives on cellular evolution could we gain by producing comparable single-cell omics data sets on strategically selected species within the plant kingdom? Previous attempts to use expression patterns of homologous genes in distantly related species to assess functional orthology have provided insights (*Adamski et al., 2020*). However, bulk tissues collected in different species may have substantially different composition of cell types (*Huang and Schiefelbein, 2015*). Furthermore, the presence of novel cell types in some species may be obscured in whole organ samples (*Kajala et al., 2021*). By focusing on the fundamental unit of organs – cell types and cell states – comparative studies of species representing key plant lineages could identify minimal molecular toolkits driving conserved plant cellular functions. In addition, these single-cell approaches could highlight the differences in developmental pathways that lead to phenotypic diversity and adaptation to a myriad of environmental conditions.

## Technical challenges faced in the development of the PCA

Critical challenges must be overcome to achieve the data generation, analysis and software development milestones of the PCA (*Figure 2*). Many mirror those faced by the larger single-cell initiatives that include animal models (*Lähnemann et al., 2020*), but we focus here on those that specifically affect the plant research community (summarized in *Box 1*).

### Isolating individual plant cells

A major challenge is dissociating plant cells: due to their rigid cell walls, some tissues and many species are recalcitrant to digestion (*Himmel et al., 2007*; *Yoo et al., 2007*). This feature limits the ability to sample mature tissues that often mediate environmental stresses. In addition, it can be difficult to isolate even young, meristematic cells in some key species that present cellular innovations. One emerging approach that has the potential to address this issue in plants is single-nucleus profiling, which avoids the need for cell wall digestion (*Giacomello, 2021*). Such nuclear isolation methods have already shown promise (*Farmer et al., 2021*; *Marand et al., 2021*). Recently developed proximity labeling approaches bypass the need for isolating cells or organelles by tagging proteins and RNAs near a target protein in living cells and organisms. These methods can profile components of protein complexes, dynamic protein-protein interactions, specific cell types and subcellular locations of proteins or RNAs (*Zhang et al., 2019a*; *Mair et al., 2019*; *Huang et al., 2020*; *Kim et al., 2019*; *Wang et al., 2019*).

### Tracking cell provenance

Another challenge is mapping cells (or nuclei) back to their tissue location in the whole plant. Methods that can rapidly and conveniently generate 'ground truth' markers of cell types in situ will be especially important as more tissues and species with few existing markers are profiled. Adapting so-called spatial transcriptomic approaches to plants will be a critical tool. In particular, untargeted methods (that is, methods based on unbiased sequencing rather than hybridization of a limited set of predefined probes) have the potential for the resolution and throughput needed (*Giacomello et al., 2017*). In addition, nomenclature systems will be necessary to link the reference maps from different species that have varying organ anatomy and specialized cell types. The hierarchical approach proposed by the common coordinate framework for humans can serve as a starting point (*Rood et al., 2019*). However, unlike the coordinate systems used in animal organ atlases which focus on accounting for differences between individuals, plant coordinate systems should instead be geared toward discovering differences among representative species and populations. Any such system should be able to accommodate differences in cell type composition (*Huang and Schiefelbein, 2015*) and the development of novel cell types (*Kajala et al., 2021*).

### Gene homology

Another set of issues surrounds gene homology in comparisons across populations and species. Plants have a well-known capacity for duplication at many scales, ranging from single gene to whole genome. An informative molecular profiling of cells will require high-quality genome annotations that consider the gene family

expansion, gene loss and complex orthology relationships which frequently occur when extensive gene duplication follows speciation. Subfunctionalization and neofunctionalization can complicate assumptions about functional orthologs. On the other hand, difficulty identifying distant homologs that share low sequence similarity could obscure functional conservation. Advancement in sequence alignment methods, such as incorporating structural information, could help to discover these distant homologs (*Morton et al., 2020*). This will probably be an iterative process, where increased resolution of gene expression at the highly resolved cellular or subcellular level could allow inferences about functional orthology across species in large gene families.

## Building the database and infrastructure of the PCA

Collection, standardization, curation, integration and visualization of data are key elements of a PCA platform, and the core PCA infrastructure should function as a data resource onto which services and tools are built.

### Experimental and analytical standards

Establishing a vetted and community-approved set of experimental and analytical standards will be important. Such an approach was key to the success of the Encyclopedia of DNA Elements project, which ensured that data from multiple international laboratories were easily compared and integrated (*ENCODE Project Consortium, 2011*). Likewise, the 1001 Arabidopsis Genomes website provided a user-friendly interface to download all sequencing data as phases were completed – allowing the community to begin analyses and discoveries even before publication – and was complemented with an interactive tool to explore phenotypes (*Seren et al., 2016*) and their genomic associations (*Togninalli et al., 2018*). The standards were made easily available and updated over time as technologies advanced.

These platforms serve as data infrastructure blueprints as well as knowledge banks and inspiration. This would allow the PCA to be extended from an atlas of a single reference species and clone, to a set of atlases capturing natural variation (within and between species) in cell types and functions. For example, the interactive PCA platform could hyperlink the gene found in each tissue or cell type to the web portal of that gene in natural variation panels of the species of interest (such as the AraGWAS Catalog, *Togninalli et al., 2020*). Data structure scaffolds in the PCA could also be created so that the template can be replicated across genotypes of a species (e.g. different ecotypes) or different species, and allow users to select their desired organism or compare between the atlases.

### Data visualization and exploration tools

In addition to data analysis and interpretation, the PCA infrastructure should include tools that provide effective ways to visualize real or virtual plant cells, and the dynamic processes that form and define them. This could enable hypothesis-generating in silico experiments that predict how plant cell types would respond to stimuli, or the properties of cells with a set of desired features. Creating these digital cells, tissues and other structures would require developing modeling, simulation, artificial intelligence, as well as data visualization and exploration tools similar to those being set up for other projects such as the OpenWorm (*Sarma et al., 2018*). Computational modeling and simulation-based predictions could then inform experimental work, the results of which could feed back into the PCA to improve models and predictions (*Radivojević et al., 2020*; *Zhang et al., 2020*).

### Curating the PCA databases

To achieve the PCA's vision of building a 4D representation of a developing plant, a wealth of existing data, resources and tools will need to be leveraged. For example, it is now routine to deposit next-generation sequencing, processed expression data, proteomics, metabolomics and protein structures in numerous repositories worldwide (*Deutsch et al., 2017*; *Kaminuma et al., 2010*; *Leinonen et al., 2011*; *NCBI Resource Coordinators, 2018*; *wwPDB consortium, 2019*). A combination of automated searches and manual curation of datasets from existing data repositories can be used to build the PCA data infrastructure. Reference genome databases such as TAIR, Phytozome, Gramene, EnsemblPlants and others that provide gene and pathway annotation resources (e.g. Gene Ontology) could enable an initial assessment of gene structure, location and function (*Berardini et al., 2015*; *Goodstein et al., 2012*; *Howe et al., 2020*; *Tello-Ruiz et al., 2021*; *The Gene Ontology Consortium, 2019*).

Building a platform that brings these data together with data exploration and visualization tools that allow a user to explore at multiple scales, from molecular to cellular to organismal, is a challenge. To this end, single-cell datasets should be made available in both raw and processed data formats and linked from the public-facing PCA website. This resource could additionally be populated using automated literature search tools, followed by manual curation by the PCA community. A selection of high-quality 'reference' single-cell datasets directly hosted by the PCA may also prove to be useful for benchmarking purposes. For all datasets, curation standards should be imposed to include minimal metadata falling into several categories: (1) experimental metadata should include a project name, details regarding how plants were grown, treated and sampled, as well as quality control metrics and definitions used to evaluate sample integrity; (2) cell metadata should include a cell identifier (where applicable), cell type annotations and other cell-level data that have come from standard or specific analysis of the dataset; and (3) gene metadata that include structural and functional annotations should be required.

In addition to providing links to data repositories, the PCA portal should include web-based tools to generate, access and query relevant data and metadata using a standardized ontology for metadata, similar to the Single Cell Expression Atlas (*Papatheodorou et al., 2020*) and the Human Cell Atlas implementation of metadata types. This will facilitate interoperability of datasets. Depending on the data type, scripts that implement standardized pipelines for single-cell data analysis could also be made available that reflect the current best practices, including those that implement methods for cross-platform and multimodal data integration as well as label transfer (e.g., *Lotfollahi et al., 2020*; *Hie et al., 2019*; *Korsunsky et al., 2019*; *Kang et al., 2021*; *Stuart et al., 2019*). Version-controlled source code that demonstrates analyses using the information in the PCA as a starting point should also be made available. These resources should promote greater use of the rapidly growing number of single-cell datasets.

Multiscale networks can provide an intuitive method to discretize and integrate different data types (*Duran-Nebreda and Bassel, 2017*). Other informatics frontiers that will be useful to explore include multi-dimensional image visualization and analysis tools such as Napari and Squidpy (*Sofroniew et al., 2021*; *Palla et al.,*

*2021*), graph-based knowledgebases for integrating datasets (e.g., Reactome; *Fabregat et al., 2018*), natural language processing for automating curation (*Braun and Lawrence-Dill, 2020*), as well as machine learning and artificial intelligence methods for integrating heterogeneous data (*Ma et al., 2020*).

### Challenges to the PCA infrastructure

Despite the widespread availability of data types described above, additional data that need to be integrated to achieve the PCA vision (metabolomics, imaging and phenotypic observations) are currently available in a variety of heterogeneous sources such as data sharing platforms (e.g. EMBL-EBI, Dryad, Figshare, and Zenodo), institutional data repositories, researcher websites or embedded within scientific publication records. A lack of centrality in data deposition makes these data less accessible.

In addition, methods for data standardization and curation will be critical tools to mitigate challenges with existing data, create best practices for collecting new data and ensure consistency to allow for data exchange and interoperability (*Sansone et al., 2012*). Experiments, datasets and other resources should be described in sufficient detail to be easily accessible and reproducible. Validating data with resource description frameworks is a means to ensure data compliance against the conceptual model it follows, data consistency and completeness. Data records and metadata should be machine-readable by using standardized vocabularies (i.e. cell type ontologies; *Bard et al., 2005*) to enable interoperability. Existing solutions and concepts of such ontologies are broadly applied in vertebrates and are available for plants, though they need to be widely adopted by the PCA community (*Avraham et al., 2008*; *Bard et al., 2005*; *Diehl et al., 2016*; *Ilic et al., 2007*; *Xia and Yanai, 2019*). Mapping ontologies to the 4D representation of the PCA will facilitate linking and visualizing the growing knowledge of plant cells.

### Originality of the PCA infrastructure

The long-term vision for the PCA infrastructure is similar in scope to other moonshot ideas such as the Human Cell Atlas (*Regev et al., 2017*), the Transparent Plant (*Henkhaus et al., 2020*) and the Virtual Plant (*Kost, 2001*). However, the PCA vision is different in several important ways. The Human Cell Atlas is for a single organism

# Box 2. The chloroplast structure, function and dynamics illustrate PCA needs.

As highlighted here, technical solutions are needed to overcome key questions and challenges in chloroplast biology. The chloroplast is an important organelle in the plant kingdom, which is responsible for harvesting and converting light into chemical energy and sequestering carbon dioxide. Whereas much is known about chloroplast form and function, many aspects of chloroplast biology remain to be elucidated, especially in the context of the 'location-to-function' paradigm and proteostasis (**Figure 7**). The chloroplast is organized into several compartments (**Solymosi et al., 2018**; **Figure 7A**). This suborganellar compartmentalization allows for highly efficient and specialized metabolism, optimized light-driven electron transport, redox reactions and more (**Rolland et al., 2018**). The function and organization of chloroplasts greatly varies across the plant lineage and cell types (**Pinard and Mizrachi, 2018**; **Figure 7B**). For example, microalgae typically have a single chloroplast with a carbon-concentrating organelle called the pyrenoid (**Barrett et al., 2021**). In plants that can perform $C_4$ photosynthesis, the chloroplasts in the bundle sheath and mesophyll cells have different compositions and functions (**Edwards et al., 2004**). A few plants even perform $C_4$ photosynthesis within individual chlorenchyma cells where $C_4$ function is obtained by spatial separation of function between dimorphic chloroplasts located in central and peripheral cytoplasmic compartments within the same cells (**Erlinghaeuser et al., 2016**; **Edwards et al., 2004**). In most land plants, the epidermal cells of leaves (except in guard cells) are devoid of chloroplasts but instead contain non-photosynthetic plastids, whereas aquatic plants are enriched in chloroplasts in their epidermal cell layer (**Larkum et al., 2017**; **Han et al., 2020**; **Figure 7C**). Recently *Arabidopsis* epidermal plastids have been shown to act in defense equipped with plant immune components (**Irieda and Takano, 2021**). Furthermore, different environmental conditions change the chloroplast proteome as well as relative positions of chloroplasts and their parts at various scales (**Kong and Wada, 2014**; **Figure 7D**). Finally, the chloroplast life cycle includes various transformations in form and function from birth to differentiation to senescence, and how these transformations occur is an understudied frontier (**Llorente et al., 2020**; **Solymosi et al., 2018**; **Figure 7E**). It is still not fully understood how chloroplast and other types of plastids differentiate, adapt, function and integrate their activities within specific cell types. Single-cell technologies bring promise for solving key challenges in cell-type and species-specific differentiation of plastids. Examples of key technologies and how they could contribute to a range of PCA-related question are listed below:

- Affinity tagging of plastids using cell-type-specific promotors driving the expression of tagged proteins that associate to the plastid surface. Recently this approach has been developed for *Arabidopsis thaliana* mitochondria (**Niehaus et al., 2020**; **Kuhnert et al., 2020**). Successful cell-type-specific plastid affinity purification would resolve molecular features, properties and associated functions of plastid with high precision. This could include determination of plastid RNA, protein and metabolite populations through high sensitivity, state of the art RNA-seq profiling and mass spectrometry techniques.

- Time-resolved plastid-targeted proximity labeling techniques to understand intra-plastid protein organization and dynamics. Recently, proximity labeling approaches have been successfully implemented in plants. They enable discovery of proteins that interact with specific regions of a bait protein, as well as identify localized protein interactions at high spatial and time-resolved resolutions (**Huang et al., 2020**; **Mair et al., 2019**; **Zhang et al., 2019b**). This novel approach would enable resolving intra-plastid protein organization and dynamics for different cell types and photosynthetic and non-photosynthetic plastids.

- Technology to detect and track retrograde signaling pathways from plastids to nuclei in response to, for example, singlet $O_2$, protein folding stress, redox and metabolites. These retrograde pathways involve a range of small molecules and likely also protein signaling cascades (**Jiang and Dehesh, 2021**; **Muñoz and Munné-Bosch, 2020**; **Pesaresi and Kim, 2019**). Non-invasive, high-resolution and high-sensitivity imaging technologies should be developed or adapted to address this challenge.

- Plastid-localized reporter systems (biosensors) to monitor pH, calcium, ATP and redox potential. There are now many such reporters (**Isoda et al., 2021**; **Okumoto and Versaw, 2017**) and a fluorescent ATP sensor protein that allows the measure of ATP concentrations in the plastid and cytosol (**Voon et al., 2018**). Functional relationships between biophysical and biochemical parameters could be probed through the systematic application of reporter systems in plastids.

- Plastid protein lifetime reporter technologies to understand cell-type-specific proteostasis. A novel protein lifetime reporter system has only been applied to address N-degron pathways in the cytosol (**Zhang et al., 2019b**). Adaption to plastids would resolve the complexities of plastid proteome remodeling in response to development and (a)biotic changes (**Bouchnak and van Wijk, 2019**).

- Synthetic biology to engineer novel plastid metabolic pathways. Plastids are excellent targets for metabolic engineering through synthetic biology: they have their own genome and generally tolerate high levels of 'foreign' protein without significantly impacting function or proteostasis (**Boehm and Bock, 2019**; **Jensen and Scharff, 2019**).

with over 200 cell types, whereas the PCA community embraces cell atlas data from a diverse array of plants. A typical plant has about 100 cell types (https://www.ebi.ac.uk/ols/ontologies/po; *Ilic et al., 2007*). Understanding the diversity and origin of cell types across the plant lineage will be a central part of the initiative, as illustrated in *Box 2* with the chloroplast as an example. The PCA is an essential step towards realizing the vision of a Transparent Plant, a simulated environment where virtual plants can be built from parts and behaviors (*Henkhaus et al., 2020*). In order to materialize such a simulation environment, where the parts are and how they work together must be understood. Finally, we see the Virtual Plant database (*Katari et al., 2010*) as an essential component of the PCA vision and infrastructure.

## Potential sources of funding for the PCA

The PCA will require adequate funding to achieve its full potential – including translational impact for agriculture (*Bartuska, 2017*; *Bol et al., 2018*; *Gök et al., 2016*; *Hu, 2020*; *Jaffe et al., 2015*; *Rosenbloom et al., 2015*). As the PCA has an international reach, programs that facilitate collaboration across borders will be valuable in advancing the goals of the initiative, such as partnering programs by the Biotechnology and Biological Sciences Research Council (BBSRC) in the United Kingdom, European Commission's Horizon Europe and the United States' Fulbright scholars program. Similar programs exist that foster alliances between scientists from the US with those in Japan, Israel and Germany; and between India and other countries. As big data, nanotechnology, computation and artificial intelligence have now become an integral part of biological research, funds to support the interdisciplinary vision of PCA will be important to foster integration of these disciplines.

While governments are the major sources of funding for basic research, their level of funding has been steadily decreasing (*Khan et al., 2020*). Foundations are the major source of private basic research funding and are less constrained by time, politics and topic. They have a high potential for funding initiatives in emerging areas such as the PCA. However, it is challenging for foundations and individuals to find worthy initiatives. To fill this gap, the Science Philanthropy Alliance was established. One of their success stories is the Chan Zuckerberg Initiative providing substantial funds to develop technologies for the Human Cell Atlas initiative. Such a partnership for the PCA would take its vision to new heights.

Industry is another potential source of funding. Over 70% of applied research and development in the US is funded by industry (*Khan et al., 2020*). A thorough understanding of plant systems is essential for designing effective agro-biotech solutions leading to new crop varieties, and to innovative crop protection products that can leverage sustainable food production. A partnership with the PCA initiative would accelerate development of new solutions for customers of the agbiotech industry. Thus, funding and collaborations, ranging over multiple disciplines and countries, and multiway interactions amongst academia, industry and philanthropy will be essential to realize the vision of the PCA.

## Conclusion and outlook

Future economies will be increasingly plant-based. Biomass is projected to be a major source of primary energy by 2050 (*Reid et al., 2020*), plant-based meat and dairy products are already transforming the food industry (*Shepon et al., 2018*), and plant-based vaccines against several diseases are increasingly being researched (reviewed by *Shahid and Daniell, 2016*). With the rapidly changing climate, economy and values in our society, now is the time to reimagine plant science as a critical component of the future not only for agriculture, but also the environment, energy, health care, manufacturing and technology.

Today, plant science constitutes a minority of life sciences in terms of funding and workforce. To meet the demands of the future economy, we need to strengthen plant science, attract talent from other fields and train the next generation of plant scientists (*Henkhaus et al., 2020*). In 2004, the evolutionary biologist John Avise wrote that ''despite nearly three decades of experience with recombinant DNA techniques, the ultimate contribution to the broader human enterprise remains uncertain'' (*Avise, 2004*). Two decades after this statement, in 2020, recombinant DNA technology ushered in the beginning of an end to a global pandemic that cost millions of lives, through a record-breaking runtime of development and deployment of mRNA and DNA-based viral vaccines. While still at its infancy, we hope that the PCA initiative has the potential to serve as a nucleator,

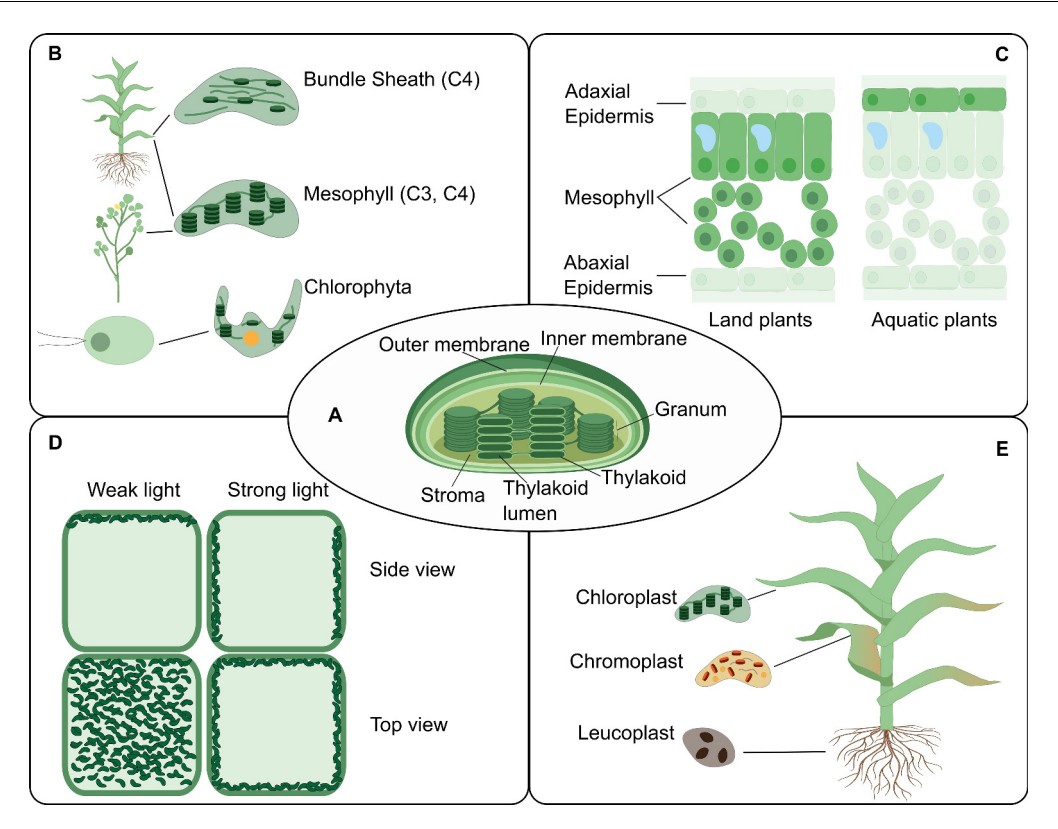

**Figure 7.** The chloroplast structure and function in connection to location. The PCA could facilitate an increased understanding of the biology of the chloroplast by providing accessible, integrative and spatially-resolved data for transcripts, proteins and metabolites across cell types (***Box 2***). (**A**) A representation of a typical chloroplast structure and its compartments. (**B**) The chloroplast structural differences in some plant lineage. (**C**) A leaf cross-section showing different chloroplast location at the tissue level in land and aquatic plants. (**D**) The chloroplast location can change within the cell upon environmental cues such as light intensity. (**E**) Various forms of chloroplasts in different organs.

bringing together scientists and engineers from a wide range of fields to solve fundamental problems in plant biology with innovative solutions from emerging technologies.

## Acknowledgements
The PCA community-building activities are funded in part by the National Science Foundation grant numbers MCB-1916797 and MCB-2052590, Carnegie Institution for Science, and BASF. We thank Emily Fryer, Nick Melosh, Heather Meyer, Jason Thomas, Terri Tippets, Renate Weizbauer, Zhiyong Wang and Kangmei Zhao for helping with organizing the first workshop. We thank Emily Fryer and Julie Gosse at the Science Editors Network for developing the PCA website. We are grateful to the first PCA workshop steering committee members Jim Haseloff, David Jackson, Edward Marcotte, John Marioni, Marisa Otegui, Alberto Salleo, Waltraud Schulze, Edgar Spalding, Michael Sussman, Marja Timmermans and HS Philip Wong for their guidance and support. We thank Rachel Shahan, Shouling Xu, Kevin Cox and Erin Zess for their input in developing the manuscript. Some images in the figures were created with BioRender.com.

### Plant Cell Atlas Consortium
**Jahed Ahmed**: Louvain Institute of Biomolecular Science and Technology, UCLouvain, Louvain La Neuve, Belgium; **Oluwafemi Alaba**: School of Biology and Ecology, University of Maine, Orono, United States; **Gazala Ameen**: Washington State University, Pullman, United States; **Vaishali Arora**: University of Delhi, Delhi, India; **Mario A Arteaga-Vazquez**: Universidad Veracruzana, Xalapa, Mexico; **Alok Arun**: Inter American University of Puerto Rico, Barranquitas, United States; **Julia Bailey-Serres**: Center for Plant Cell Biology and Department of Botany and Plant Sciences, University of California, Riverside, Riverside, United

States; **Laura E Bartley**: Washington State University, Pullman, United States; **George W Bassel**: University of Warwick, Coventry, United Kingdom; **Dominique C Bergmann**: Stanford University and Howard Hughes Medical Institute, Stanford, United States; **Edoardo Bertolini**: Donald Danforth Plant Science Center, St. Louis, United States; **Kaushal Kumar Bhati**: Louvain Institute of Biomolecular Sciences, Catholic University of Louvain, Louvain La Neuve, Belgium; **Noel Blanco-Touriñán**: Plant Vascular Development Group, ETH Zürich, Zurich, Switzerland; **Steven P Briggs**: University of California San Diego, La Jolla, United States; **Javier Brumos**: Instituto de Biologia Molecular y Celular de Plantas (IBMCP) Consejo Superior de Investigaciones Científicas (CSIC), Valencia, Spain; **Benjamin Buer**: Bayer AG, Monheim am Rhein, Germany; **Adrien Burlaocot**: HHMI, Department of Plant and Microbial Biology, University of California, Berkeley, Berkeley, United States; **Sergio Alan Cervantes-Pérez**: LANGEBIO-CINVESTAV, Irapuato, Mexico; **Sixue Chen**: Department of Biology, University of Florida, Gainesville, United States; **Bruno Contreras-Moreira**: European Bioinformatics Institute - EMBL, Hinxton, United Kingdom; **Francisco J CORPAS**: Estación Experimental del Zaidin (CSIC), Granada, Spain; **Alfredo Cruz-Ramirez**: National Laboratory of Genomics for Biodiversity CINVESTAV, Irapuato, Mexico; **Cesar L Cuevas-Velazquez**: Departamento de Bioquímica, Facultad de Química, Universidad Nacional Autónoma de México, Cd., Irapuato, Mexico; **Josh T Cuperus**: University of Washington, Seattle, United States; **Lisa I David**: North Carolina State University, Raleigh, United States; **Stefan de Folter**: Unidad de Genómica Avanzada (UGA-LANGEBIO), Centro de Investigación y de Estudios Avanzados del Instituto Politécnico Nacional (CINVESTAV-IPN), Irapuato, Mexico; **Peter H Denolf**: BASF Innovation Center Gent, Zwijnaarde, Belgium; **Pingtao Ding**: Institute of Biology Leiden, Leiden University, Leiden, Netherlands; **William P Dwyer**: Department of Plant Biology, Carnegie Institution for Science, Stanford, United States; **Matthew MS Evans**: Department of Plant Biology, Carnegie Institution for Science, Stanford, United States; **Nancy George**: European Bioinformatics Institute, Cambridge, United Kingdom; **Pubudu P Handakumbura**: Environmental Molecular Sciences Division, Pacific Northwest National Laboratory, Richland, United States; **Maria J Harrison**: Boyce Thompson Institute, Ithaca, United States; **Elizabeth S Haswell**: Washington University in St. Louis, St. Louis, United States; **Venura Herath**: University of Peradeniya, Peradeniya, Sri Lanka; **Yuling Jiao**: Institute of Genetics and Developmental Biology, Chinese Academy of Sciences, Beijing, China; **Robert E Jinkerson**: University of California, Riverside, Riverside, United States; **Uwe John**: Alfrd-Wegener-Institute Helmholtz centre for Polar and Marine Research, Bremerhaven, Germany; **Sanjay Joshi**: University of Kentucky, Lexington, United States; **Abhishek Joshi**: Mohanlal Sukhadia University, Udaipur, India; **Lydia-Marie Joubert**: Stanford University, Stanford, United States; **Ramesh Katam**: Florida A&M University, Tallahassee, United States; **Harmanpreet Kaur**: Punjab Agricultural University, Ludhiana, India; **Yana Kazachkova**: Weizmann Institute of Science, Rehovot, Israel; **Sunil K Kenchanmane Raju**: Michigan State University, East Lansing, United States; **Mather A Khan**: Heinrich Heine University, Dusseldorf, Germany; **Rajdeep Khangura**: Purdue University, West Lafayette, United States; **Ajay Kumar**: ICAR-NIPB, New Delhi, India; **Arun Kumar**: Biotechnology Division, CSIR-Institute of Himalayan Bioresource Technology, Palampur, India; **Pankaj Kumar**: Dr Yashwant Singh Parmar University of Horticulture and Forestry Nauni Solan Himachal Pradesh, Solan, India; **Pradeep Kumar**: Centre for Cellular & Molecular Biology, Hyderabad, India; **Dhruv Lavania**: University of Alberta, Edmonton, Canada; **Tedrick Thomas Salim Lew**: Institute of Materials Research and Engineering, Singapore, Singapore; **Mathew G Lewsey**: La Trobe University, Bundoora, Australia; **Chien-Yuan Lin**: Joint BioEnergy Institute, Lawrence Berkeley National Laboratory, Emeryville, United States; **Dianyi Liu**: Donald Danforth Plant Science Center, Saint Louis, United States; **Le Liu**: Department of Biology, University of Massachusetts Amherst, Amherst, United States; **Tie Liu**: University of Florida, Gainesville, United States; **Ansul Lokdarshi**: University of Tennessee, Knoxville, United States; **Ai My Luong**: Louvain Institute of Biomolecular Science and Technology, Louvain-la-Neuve, Belgium; **Iain C Macaulay**: Earlham Institute, Norwich, United Kingdom; **Sakil Mahmud**: Plant Cell Biology, IZMB, University of Bonn, Bonn, Germany; **Ari Pekka Mähönen**: Institute of Biotechnology, HiLIFE, University of Helsinki, Helsinki, Finland; **Kamal Kumar Malukani**: Molecular Biology, Hyderabad, India; **Alexandre P Marand**: University of Georgia, Athens, United States; **Carly A Martin**: MIT, Cambridge, United States; **Claire D McWhite**: Princeton University, Princeton, United States; **Devang Mehta**: University of Alberta, Edmonton, Canada; **Miguel Miñambres Martín**: Heinrich Heine University, Düsseldorf, Germany; **Jenny C Mortimer**: University of Adelaide, Adelaide, Australia; **Lachezar A Nikolov**: University of California, Los Angeles, Los, Angeles, United States; **Tatsuya Nobori**: Salk Institute for Biological Studies, San, Diego, United States; **Trevor M Nolan**: Duke University, Durham, United States; **Aaron J Ogden**: Pacific Northwest National Laboratory, Richland, United States; **Marisa S Otegui**: University of Wisconsin-Madison, Madison, United States; **Mark-Christoph Ott**: Bayer AG, Monheim am Rhein, Germany; **José M Palma**: Estación Experimental del Zaidín, CSIC, Granada, Spain; **Puneet Paul**: University of Nebraska-Lincoln, Lincoln, United States; **Atique U Rehman**: Department of Agronomy, Bahauddin Zakariya University Multan, Multan, Pakistan; **Maida Romera-Branchat**: Institute of Plant Biology and Biotechnology, University of Münster, Muenster (Germany), Germany; **Luis C Romero**: Instituto de Bioquimica Vegetal y Fotosintesis, CSIC-US, Seville, Spain; **Ronelle Roth**: Department of Plant Sciences, University of Oxford, Oxford, United Kingdom; **Saroj**

**K Sah**: The Pennsylvania State University, University Park, United States; **Rachel Shahan**: Duke University, Durham, United States; **Shyam Solanki**: Washington State University, Pullman, United States; **Bao-Hua Song**: University of North Carolina at Charlotte, Charlotte, United States; **Rosangela Sozzani**: North Carolina State University, Raleigh, United States; **Gary Stacey**: University of Missouri, Columbia, United States; **Anna N Stepanova**: North Carolina State University, Raleigh, United States; **Nicolas L Taylor**: The University of Western Australia, Perth, Australia; **Marcela K Tello-Ruiz**: Cold Spring Harbor Laboratory, Cold Spring Harbor, United States; **Tuan M Tran**: Nanyang Technological University, Singapore, Singapore; **Rajiv Kumar Tripathi**: University of Manitoba, Winnipeg, Canada; **Batthula Vijaya Lakshmi Vadde Vadde**: Cornell University, Ithaca, United States; **Tamas Varga**: Environmental Molecular Sciences Division, Pacific Northwest National Laboratory, (PNNL), Richland, United States; **Marija Vidovic**: Institute of Molecular Genetics and Genetic Engineering, University of Belgrade, Belgrade, Serbia; **Justin W Walley**: Iowa State University, Ames, United States; **Zhiyong Wang**: Carnegie Institution for Science, Dept. of Plant Biology, Stanford, United States; **Renate A Weizbauer**: Carnegie Institution for Science, Dept. of Plant Biology, Stanford, United States; **James Whelan**: La Trobe University, Melbourne, Australia; **Asela J Wijeratne**: Arkansas State University, Jonesboro, United States; **Tingting Xiang**: University of North Carolina at Charlotte, Charlotte, United States; **Shouling Xu**: Carnegie Institution for Science, Stanford, United States; **Ramin Yadegari**: University of Arizona, Tucson, United States; **Houlin Yu**: Department of Biochemistry and Molecular Biology, University of Massachusetts Amherst, Amherst, United States; **Hai Ying Yuan**: University of Saskatchewan, Saskatoon, Canada; **Fabio Zanini**: UNSW, Sydney, Australia; **Feng Zhao**: Laboratoire Reproduction et Développement des Plantes, Université de Lyon, Lyon, France; **Jie Zhu**: University of California, Davis, Davis, United States; **Xiaohong Zhuang**: Centre for Cell and Developmental Biology, State Key Laboratory of Agrobiotechnology, School of Life Sciences, The Chinese University of Hong Kong, Hong Kong SAR, Shatin, Hong Kong;

**Suryatapa Ghosh Jha** is in the Department of Plant Biology, Carnegie Institution for Science, Stanford, United States

https://orcid.org/0000-0001-7976-1272

**Alexander T Borowsky** is in the Department of Botany and Plant Sciences, University of California, Riverside, Riverside, United States

https://orcid.org/0000-0002-2510-8136

**Benjamin J Cole** is in the Joint Genome Institute, Lawrence Berkeley National Laboratory, Walnut Creek, United States

https://orcid.org/0000-0001-9652-624X

**Noah Fahlgren** is in the Donald Danforth Plant Science Center, St. Louis, United States

https://orcid.org/0000-0002-5597-4537

**Andrew Farmer** is in the National Center for Genome Resources, Santa Fe, United States

https://orcid.org/0000-0002-4224-2433

**Shao-shan Carol Huang** is in the Center for Genomics and Systems Biology, New York University, New York, United States

**Purva Karia** is in the Department of Plant Biology, Carnegie Institution for Science, Stanford, United States and the Department of Cell and Systems Biology, University of Toronto, Toronto, Canada

https://orcid.org/0000-0002-4263-4867

**Marc Libault** is in the Department of Agronomy and Horticulture, University of Nebraska-Lincoln, Lincoln, United States

https://orcid.org/0000-0001-7419-9129

**Nicholas J Provart** is in the Department of Cell and Systems Biology and the Centre for the Analysis of Genome Evolution and Function, University of Toronto, Toronto, Canada

https://orcid.org/0000-0001-5551-7232

**Selena L Rice** is in the Department of Plant Biology, Carnegie Institution for Science, Stanford, United States

https://orcid.org/0000-0001-8403-5785

**Maite Saura-Sanchez** is in the Consejo Nacional de Investigaciones Científicas y Técnicas, Instituto de Investigaciones Fisiológicas y Ecológicas Vinculadas a la Agricultura, Facultad de Agronomía, Universidad de Buenos Aires, Buenos Aires, Argentina

https://orcid.org/0000-0001-8115-3540

**Pinky Agarwal** is in the National Institute of Plant Genome Research, New Delhi, India

**Amir H Ahkami** is in the Environmental Molecular Sciences Division, Pacific Northwest National Laboratory, Richland, United States

https://orcid.org/0000-0002-0545-1236

**Christopher R Anderton** is in the Environmental Molecular Sciences Division, Pacific Northwest National Laboratory, Richland, United States

**Steven P Briggs** is in the Department of Biological Sciences, University of California, San Diego, United States

https://orcid.org/0000-0002-7226-8618

**Jennifer AN Brophy** is in the Department of Biology, Stanford University, Stanford, United States

https://orcid.org/0000-0001-7808-4281

**Peter Denolf** is in the BASF Seeds & Traits, Ghent, Belgium

https://orcid.org/0000-0003-4567-2700

**Luigi F Di Costanzo** is in the Department of Agricultural Sciences, University of Naples Federico II, Napoli, Italy

https://orcid.org/0000-0002-4795-2573

**Moises Exposito-Alonso** is in the Department of Plant Biology, Carnegie Institution for Science, Stanford, United States

https://orcid.org/0000-0001-5711-0700

**Stefania Giacomello** is in the SciLifeLab, KTH Royal Institute of Technology, Solna, Sweden

https://orcid.org/0000-0003-0738-1574

**Fabio Gomez-Cano** is in the Department of Biochemistry and Molecular Biology, Michigan State University, East Lansing, United States

https://orcid.org/0000-0002-2624-0112

**Kerstin Kaufmann** is in the Department for Plant Cell and Molecular Biology, Institute for Biology, Humboldt-Universitaet zu Berlin, Berlin, Germany

https://orcid.org/0000-0001-7960-6256

**Dae Kwan Ko** is in the Great Lakes Bioenergy Research Center, Michigan State University, East Lansing, United States

https://orcid.org/0000-0002-9720-5138

**Sagar Kumar** is in the Department of Plant Breeding & Genetics, Mata Gujri College, Fatehgarh Sahib, Punjabi University, Patiala, India

https://orcid.org/0000-0002-0725-0059

**Andrey V Malkovskiy** is in the Department of Plant Biology, Carnegie Institution for Science, Stanford, United States

https://orcid.org/0000-0002-5648-8602

**Naomi Nakayama** is in the Department of Bioengineering, Imperial College London, London, United Kingdom

**Toshihiro Obata** is in the Department of Biochemistry, University of Nebraska-Lincoln, Lincoln, United States

https://orcid.org/0000-0001-8931-7722

**Marisa S Otegui** is in the Department of Botany, University of Wisconsin-Madison, Madison, United States

https://orcid.org/0000-0003-4699-6950

**Gergo Palfalvi** is in the Division of Evolutionary Biology, National Institute for Basic Biology, Okazaki, Japan

**Elsa H Quezada-Rodríguez** is in the Ciencias Agrogenómicas, Escuela Nacional de Estudios Superiores Unidad León, Universidad Nacional Autónoma de México, León, Mexico

https://orcid.org/0000-0001-7789-4987

**Rajveer Singh** is in the School of Agricultural Biotechnology, Punjab Agricultural University, Ludhiana, India

https://orcid.org/0000-0002-2922-1371

**R Glen Uhrig** is in the Department of Science, University of Alberta, Edmonton, Canada

https://orcid.org/0000-0003-2773-4381

**Jamie Waese** is in the Department of Cell and Systems Biology/Centre for the Analysis of Genome Evolution and Function, University of Toronto, Toronto, Canada

https://orcid.org/0000-0001-8783-9914

**Klaas Van Wijk** is in the School of Integrated Plant Science, Plant Biology Section, Cornell University, Ithaca, United States

https://orcid.org/0000-0001-9536-0487

**R Clay Wright** is in the Department of Biological Systems Engineering, Virginia Tech, Blacksburg, United States

**David W Ehrhardt** is in the Department of Plant Biology, Carnegie Institution for Science, Stanford, United States

dehrhardt@carnegiescience.edu

**Kenneth D Birnbaum** is in the Center for Genomics and Systems Biology, New York University, New York, United States

kdb5@nyu.edu

**Seung Y Rhee** is in the Department of Plant Biology, Carnegie Institution for Science, Stanford, United States

srhee@carnegiescience.edu

https://orcid.org/0000-0002-7572-4762

*Author contributions:* Plant Cell Atlas Consortium, Moises Exposito-Alonso, Klaas Van Wijk, Writing - review and editing; Suryatapa Ghosh Jha, Noah Fahlgren, Andrew Farmer, Shao-shan Carol Huang, Nicholas J Provart, Toshihiro Obata, Supervision, Writing - original draft, Writing - review and editing; Alexander T Borowsky, Supervision, Visualization, Writing - original draft, Writing - review and editing; Benjamin J Cole, Pinky Agarwal, Amir H Ahkami, Christopher R Anderton, Steven P Briggs, Jennifer AN Brophy, Peter Denolf, Luigi F Di Costanzo, Stefania Giacomello, Fabio Gomez-Cano, Kerstin Kaufmann, Dae Kwan Ko, Sagar Kumar, Naomi Nakayama, Marisa S Otegui, Gergo Palfalvi, Elsa H Quezada-Rodríguez, Rajveer Singh, R Glen Uhrig, R Clay Wright, Writing - original draft, Writing - review and editing; Purva Karia, Marc Libault, Maite Saura-Sanchez, Andrey V Malkovskiy, Visualization, Writing - original draft, Writing - review and editing; Selena L Rice, Supervision, Writing - original draft, Project administration, Writing - review and editing; Jamie Waese, Visualization, Writing - review and editing; David W Ehrhardt, Kenneth D Birnbaum, Conceptualization, Funding acquisition, Writing - original draft, Writing - review and editing; Seung Y Rhee, Conceptualization, Supervision, Funding acquisition, Writing - original draft, Project administration, Writing - review and editing

*Competing interests:* The authors declare that no competing interests exist.

## Funding

| Funder | Grant reference number | Author |
| --- | --- | --- |
| National Science Foundation | 1916797 | David W Ehrhardt Kenneth D Birnbaum Seung Yon Rhee |
| National Science Foundation | 2052590 | Seung Yon Rhee |

The funders had no role in study design, data collection and interpretation, or the decision to submit the work for publication.

## Decision letter and Author response

Decision letter https://doi.org/10.7554/eLife.66877.sa1
Author response https://doi.org/10.7554/eLife.66877.sa2

## Additional files

### Data availability

No associated datasets.

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
