## [Decision Letter]

Thank you for submitting your article "A Roadmap for the Plant Cell Atlas" to *eLife* for consideration as a Feature Article. Three peer reviewers have reviewed your article, and two editors from the *eLife* Features Team have overseen the evaluation.

The reviewers and editors have discussed the reviews and we have drafted this decision letter to help you prepare a revised submission.

Summary:

On the heels of a 2020 workshop, this paper discusses the Plant Cell Atlas (PCA) initiative, which aims to create a resource to identify all plant cell types, and to annotate the localization and organization of molecules at cellular and tissue levels. The authors discuss the gaps in knowledge and technologies and the challenges of integrating different data types to enable the PCA infrastructure. This vision paper provides a compelling argument as to why the PCA is important for food security, environmental sustenance, and human health. However, a number of points need to be addressed to make the article suitable for publication.

Overall, the reviewers raised concerns about two aspects: ensuring that the paper is markedly different from the one published in 2019; and clarifying whether the manuscript is a clarion call for investment in plant science or a highly technical discussion of data collection at a finer and finer level. More specifically, you will find below a list of essential revisions for you to consider.

Essential revisions:

1. Overall: Please include a detailed, practical roadmap of how the PCA can come to be in the near future, which would include concrete steps of how it could be designed and rolled out. Without including this content, the reviewers fear the paper would be too similar to previous work and would lack the impact it could otherwise achieve on the community. This feedback constitutes one of the most important concerns from the reviewers.

2. Overall: Please address how the PCA consortium plans to tackle how to unify the growing number of single-cell data, and make them more accessible/comparable to the community. For instance, please consider including standardized techniques, pipelines and SOPs to maximize productive sharing and cross-comparison/analysis of these data. The reviewer is especially concerned about the fact that ever accumulating scRNA-seq (and other single-cell) data are under-utilized because it is very difficult, if not impossible, to compare different technological platforms. Indeed, different pipelines for dimension reduction/data presentation make it difficult to cross-analyze data generated from different laboratories in a meaningful way, and different genetic backgrounds, growth conditions, age, stress treatment, etc, make a comparison of Arabidopsis data even more challenging. This discussion could also include reflection on how to make single-cell data comparable to metazoan data.

3. Overall: Please consider strengthening your discussion of the roadblocks particular to plants (as opposed to other systems such as metazoans or humans) when leveraging experimental and computational technologies to create the atlas, and in particular how the data will actually be integrated and used.

4. Overall: Please also discuss more clearly how piecing together these different sources of data will result in a new, usable structure. The reviewer felt that discussing these issues was 'lost' amongst the text and could be made more prominent.

5. Overall: As natural variation is important in determining phenotypic diversity, please clarify how the PCA initiative is going to handle such complexity in terms of extending the functionality of cell atlas beyond one genotype.

6. Title: Please change the title of the article to reflect that it will not detail a prototype of PCA infrastructure, and to reflect that it is more a collection of focus areas or opportunities. Please ensure that this title is markedly different from the previous published work "Towards building a plant cell atlas (2019)".

7. Page 4, start of the page: Please discuss how PCA will improve on what is currently being done, as the benefits of understanding cell/tissue-specific expression have been known – and used – for decades.

8. Page 4, in "vision for a plant cell atlas": If possible, please describe other examples or prototypes in model organisms such as human, mouse or *Drosophila*, which would support that it is feasible for the PCA community to accomplish its vision in the ways you suggest (e.g. example creating "a one-stop, user-friendly website", "a 4D representation of a developing plant, from root to shoot with data collected from single-cell omics platforms for each cell".)

9. Page 7: Please consider trying to aggregate the gaps for investment, which at present are scattered through the review and sometimes not fully discussed. For instance, "Gaps in technology" is a comprehensive review of new techniques, but only seems to identify "more efficient capture technologies" as a specific gap.

10. End of page 12: Please consider further discussing and expanding on how the PCA compares to the roadmap of other projects and ideas, such as the Human Cell Atlas, and the actual concepts in the Decadal Vision ("The Transparent Plant"), which, while referenced, are not considered even though they outline similar issues.

11. End of page 12: Continuing this line of thoughts, please discuss how the PCA Consortium will/does coordinate with existing platforms to deliver the vision. For instance, the plant biology community already widely uses numerous data-curated plant atlas platforms such as BAR (http://bar.utoronto.ca/), ATTED-II (www.atted.jp/). Other examples include the human cell atlas project (https://www.humancellatlas.org/). It would be enlightening if the authors could discuss whether the PCA will stay separate and become yet another platform, or if it will be possible to actively coordinate and integrate the PCA platform with these existing platforms. What platform would be the most accessible to a variety of users (from students to senior researchers, from bioinformaticians to agricultural specialists)?

12. Page 12 and 13: Please add content to explain in more detail what will happen to bring the 4D project to life. For instance, please discuss the informatics frontiers and how we get to “hypothesis-generating in silico experiments" that could potentially meet some of the "people and culture objectives”.

13. Page 13: Please consider fleshing out the "people and culture" aspect beyond the two existing paragraphs.

---

## [Author Response]

Overall, the reviewers raised concerns about two aspects: ensuring that the paper is markedly different from the one published in 2019; and clarifying whether the manuscript is a clarion call for investment in plant science or a highly technical discussion of data collection at a finer and finer level. More specifically, you will find below a list of essential revisions for you to consider.

We thank the reviewers and editors for calling us out on this important point. We wish the paper to be “a clarion call for investment in plant science” and, in the revised version, we highlight both the opportunities and challenges in making this happen from technical, human resource, and investment perspectives. We have substantially revised the manuscript to provide evidence and argument supporting this primary message and detailed the mechanics of how we envision building out this important resource for plant science. We added 3 new figures and 2 boxes to address the reviewers’ and editors’ comments. We provide a detailed point-by-point response below that fully addresses the comments. Overall, we feel that the comments from the editors and reviewers guided us well in devising a much stronger and more impactful perspective article for which we are grateful.

Essential revisions:1. Overall: Please include a detailed, practical roadmap of how the PCA can come to be in the near future, which would include concrete steps of how it could be designed and rolled out. Without including this content, the reviewers fear the paper would be too similar to previous work and would lack the impact it could otherwise achieve on the community. This feedback constitutes one of the most important concerns from the reviewers.

We substantially revised the manuscript, emphasizing how the PCA infrastructure can be built as a community-supported, integrated resource for assimilating molecular and cellular data to make accessible the dynamic organization of various plant cell types. First, we laid out a pragmatic set of achievable targets for the next 10+ years in the “Vision for a Plant Cell Atlas” section, and summarized the milestones in Figure 2 (new).

Second, we added a graphical illustration of a “one-stop shop” PCA resource in Figure 3 (new) that exemplifies the type of data integration and visualization we are envisioning.

Third, throughout the manuscript, we added details on how the PCA could be built. These include data integration approaches (“Vision for the Plant Cell Atlas” section); community building approaches (“The PCA Community” section); types of infrastructure needed for making single cell data accessible according to the FAIR principles (“Data and Infrastructure” section); and specific challenges that need to be overcome (“Challenges Faced in the Development of the PCA” section). All the new text are in blue.

Fourth, we summarized the major challenges that should be targeted for investment to enable the PCA and transform plant science in Box 1.

Finally, we added an example of plant biology, the chloroplast structure, function and dynamics, for which PCA efforts could transform its understanding and impact in plant science in Box 2 and associated Figure 7 (new).

2. Overall: Please address how the PCA consortium plans to tackle how to unify the growing number of single-cell data, and make them more accessible/comparable to the community. For instance, please consider including standardized techniques, pipelines and SOPs to maximize productive sharing and cross-comparison/analysis of these data. The reviewer is especially concerned about the fact that ever accumulating scRNA-seq (and other single-cell) data are under-utilized because it is very difficult, if not impossible, to compare different technological platforms. Indeed, different pipelines for dimension reduction/data presentation make it difficult to cross-analyze data generated from different laboratories in a meaningful way, and different genetic backgrounds, growth conditions, age, stress treatment, etc, make a comparison of Arabidopsis data even more challenging. This discussion could also include reflection on how to make single-cell data comparable to metazoan data.

To address how the PCA will facilitate access and use of the single cell data sets, we have laid out in more detail our vision of single-cell dataset accessibility in the ‘Data and Infrastructure’ section by adding a new paragraph.

3. Overall: Please consider strengthening your discussion of the roadblocks particular to plants (as opposed to other systems such as metazoans or humans) when leveraging experimental and computational technologies to create the atlas, and in particular how the data will actually be integrated and used.

We have included a description of challenges in a new section called “Challenges Faced in the Development of the PCA” that replaces the old section on “Gaps in Technology” discusses the challenges as well as ways to overcome those in the future. We also added a summary of key challenges in Box 1 that are particular to the goals of the PCA such as natural variation and interspecies comparison, polyploidy and multigene families, spatial constraints imposed by cell walls, differences between tissues in terms of accessibility, and the limitation of resources in the plant science community.

4. Overall: Please also discuss more clearly how piecing together these different sources of data will result in a new, usable structure. The reviewer felt that discussing these issues was 'lost' amongst the text and could be made more prominent.

We have addressed this by clarifying the vision statement. We have clarified what the benefits are for bringing all the pieces together and the revised manuscript now emphasizes integration, usability, and visualization in the “Vision for Plant Cell Atlas” section. We have also included a new figure (Figure 3) showing an integrated user interface that exemplifies the type of front-end that is made possible by the back end of integrated data we are envisioning.

5. Overall: As natural variation is important in determining phenotypic diversity, please clarify how the PCA initiative is going to handle such complexity in terms of extending the functionality of cell atlas beyond one genotype.

We agree with the reviewer that it is important to plan for natural variation to be incorporated in the PCA. We have incorporated references to genomic natural variation databases such as the 1001 Arabidopsis Genomes in the “Data and Infrastructure” section. We envision several degrees of integration of natural variation in the PCA, from hyper-linking gene IDs between PCA and natural variation catalogs to the preparation of data structure scaffolds that will enable in the future re-creation of PCA for multiple genotypes and species and interactively compare and study differences and similarities.

6. Title: Please change the title of the article to reflect that it will not detail a prototype of PCA infrastructure, and to reflect that it is more a collection of focus areas or opportunities. Please ensure that this title is markedly different from the previous published work "Towards building a plant cell atlas (2019)".

We changed the title to: “The Plant Cell Atlas”

[Editor’s note: The title has now been changed following discussions with the editor.]

7. Page 4, start of the page: Please discuss how PCA will improve on what is currently being done, as the benefits of understanding cell/tissue-specific expression have been known – and used – for decades.

We have addressed how PCA will improve the accessibility of tissue and single cell data and eliminate the constraint in using transcriptomic data in comparative analysis studies. The PCA community as well as the portal will be useful as it will promote community supported best practices in data generation and use and will also integrate metrics and reference maps for organ/cell types. This is elaborated in the “Vision for a Plant Cell Atlas” section.

8. Page 4, in "vision for a plant cell atlas": If possible, please describe other examples or prototypes in model organisms such as human, mouse or *Drosophila*, which would support that it is feasible for the PCA community to accomplish its vision in the ways you suggest (e.g. example creating "a one-stop, user-friendly website", "a 4D representation of a developing plant, from root to shoot with data collected from single-cell omics platforms for each cell".)

We added a paragraph to the “Vision for a Plant Cell Atlas” section to address this point.

9. Page 7: Please consider trying to aggregate the gaps for investment, which at present are scattered through the review and sometimes not fully discussed. For instance, "Gaps in technology" is a comprehensive review of new techniques, but only seems to identify "more efficient capture technologies" as a specific gap.

We have addressed this comment by including a revised section titled “Challenges Faced in the Development of PCA” to consolidate the limitations and describes the areas that should be targeted for investment in the future. This section is summarized in Box 2.

10. End of page 12: Please consider further discussing and expanding on how the PCA compares to the roadmap of other projects and ideas, such as the Human Cell Atlas, and the actual concepts in the Decadal Vision ("The Transparent Plant"), which, while referenced, are not considered even though they outline similar issues.

We added a paragraph in the “Data and Infrastructure” section to address this point.

11. End of page 12: Continuing this line of thoughts, please discuss how the PCA Consortium will/does coordinate with existing platforms to deliver the vision. For instance, the plant biology community already widely uses numerous data-curated plant atlas platforms such as BAR (http://bar.utoronto.ca/), ATTED-II (www.atted.jp/). Other examples include the human cell atlas project (https://www.humancellatlas.org/). It would be enlightening if the authors could discuss whether the PCA will stay separate and become yet another platform, or if it will be possible to actively coordinate and integrate the PCA platform with these existing platforms. What platform would be the most accessible to a variety of users (from students to senior researchers, from bioinformaticians to agricultural specialists)?

To address this comment, we overhauled the section entitled “The PCA Community” and added a new subsection called “PCA in the bioinformatics community” where we lay out how we see PCA positioning in itself in the context of the other existing resources.

12. Page 12 and 13: Please add content to explain in more detail what will happen to bring the 4D project to life. For instance, please discuss the informatics frontiers and how we get to hypothesis-generating in silico experiments" (p. 12) that could potentially meet some of the "people and culture objectives (p. 13).

We added examples of informatics areas we envision being important for realizing the PCA vision to the Data and Infrastructure section. First, we expanded the need for modeling, simulation, and visualization tools for developing a virtual plant cell resource, which would be conceptually similar to other digital organism projects like the OpenWorm. Second, we introduce several topic areas with references that will likely be important informatics frontiers for the PCA, including multi-dimensional image visualization and analysis, graph-based knowledge bases, natural language processing, and machine learning and artificial intelligence.

13. Page 13: Please consider fleshing out the "people and culture" aspect beyond the two existing paragraphs.

We have overhauled this section that is now called “The PCA community” and has four sections, “Developers”, “Users”, “PCA in the bioinformatics community”, and “Training”. This section has been expanded and moved up in the manuscript.